# Identification of PANoptosis-associated genes in hepatic ischemia-reperfusion injury by integrated bioinformatics analysis and machine learning

Alimu Tulahong[1☯], Xinlu Xu[1☯], Aimitaji Abulaiti[1☯], Talaiti Tuergan[1], Pingping Qiao[1], Dalong Zhu[1], Chang Liu[1], Rexiati Ruze[1], Tuerganaili Aji[1*], Yingmei Shao[1,2*]

1 Hepatobiliary and Echinococcosis Surgery Department, Digestive and Vascular Surgery Center, First Affiliated Hospital of Xinjiang Medical University, Urumqi, China, 2 State Key Laboratory of Pathogenesis, Prevention and Treatment of High Incidence Diseases in Central Asia, Urumqi, China

☯ Alimu Tulahong, Xinlu Xu and Aimitaji Abulaiti made equivalent contributions to this research as co-first authors.
* syingmei1@163.com (YS), tuergan78@sina.com (TA)

## Abstract

### Background

In the context of liver resection and transplantation, hepatic ischemia-reperfusion injury (hepatic IRI) remains a significant clinical challenge, profoundly impacting both postoperative short- and long-term recovery. A novel cell death pathway, PANoptosis, is implicated in infections, malignancies and sterile inflammation. While the specific role of PANoptosis in the development of hepatic IRI remains unclear, this study aims to provide new insights and perspectives into the underlying mechanisms, thereby addressing this knowledge gap.

### Methods

We constructed a panoptosis-related gene panel and analyzed seven gene expression datasets on hepatic IRI available in the GEO database. Differential expression analysis, enrichment analysis, and multi-omics consensus analysis were performed on panoptosis-hepatic IRI DEGs. Core genes associated with hepatic IRI were identified using ten machine learning algorithms. Single-cell analysis and two immune infiltration algorithms were employed to assess immune cell infiltration and their interplay with core genes. To validate our findings, core gene expression was validated via serum detection, H&E staining, and quantitative real-time PCR in a mouse hepatic IRI.

### Results

We identified 52 DE-PANRGs from a constructed panoptosis-related gene set of 485 genes. Functional enrichment analysis indicated their participation in necroptosis,

**Data availability statement:** Seven hepatic IRI-associated datasets were retrieved from GEO (Gene Expression Omnibus, https://www.ncbi.nlm.nih.gov/geo/): GSE151648, GSE12720, GSE15480, GSE23649, GSE87487, GSE112713, and GSE189539.

**Funding:** This work was supported by the NSFC No.82360111; Xinjiang Science and Technology Department—Leading talents in technological innovation - high-level leading talents NO.2022TSYCLJ0034; State Key Laboratory for the Cause and Control of High Incidence in Central Asia Jointly Constructed by the Ministry and the Province NO. SKL-HIDCA-2023-2; and Xinjiang Uygur Autonomous Region Graduate Innovation Program, No. XJ2024G153. Funding was provided to support the study design, the successful execution of the experiments, the purchase of experimental materials, and the payment of article publication fees. The authors associated with these grants are Tuerganaili Aji and Yingmei Shao. The funders had no role in data collection and analysis, decision to publish, or preparation of the manuscript. There was no additional external funding received for this study.

**Competing interests:** The authors have declared that no competing interests exist.

apoptosis, cytokine-cytokine receptor interaction, NOD-like receptor signaling, and other related processes. Three distinct molecular subtypes of hepatic IRI were identified, with subtype C2 showing high expression of DE-PANRGs. Machine learning identified eight feature genes (IER3, CDKN1A, EMP1, IL1B, BTG3, JUN, HSPB1, and IL1A) with diagnostic potential. The function and correlation of core genes were confirmed through single-cell and immune infiltration analyses, and validated in a mouse hepatic IRI model.

## Conclusion

Utilizing a panoptosis gene set, this study identified eight core genes involved in hepatic IRI, providing novel insights into panoptosis' role in hepatic IRI.

---

## Introduction

Hepatic ischemia-reperfusion injury (hepatic IRI) commonly occurs during major hepatectomy, liver transplantation and autotransplantation, trauma, and shock. Restoration of blood flow after ischemia paradoxically worsens hepatocellular injury and dysfunction, particularly following liver transplantation [1,2]. The inflammatory response in IRI increases short-term risks, including acute rejection, early allograft dysfunction or primary non-function, liver failure, and even multi-organ dysfunction [3–5]. IRI is also associated with long-term adverse outcomes—chronic rejection, impaired regenerative capacity, biliary complications, cancer recurrence, and fibrosis, that significantly affect transplant outcomes [6]. Multiple pathways contribute to the pathogenesis of hepatic IRI, including oxidative stress, Calcium Overload, inflammatory responses, endothelial cell injury, complement activation, and cell death [7]; Nevertheless, the existing intervention and pharmacological strategies fail to fully prevent liver graft IRI. A deeper understanding of the inflammatory mechanisms of hepatic IRI is essential for developing therapies that minimize acute graft injury and reduce disease recurrence in liver transplant recipients. Targeting acute inflammation may help restore early graft function but also lower the incidence or severity of subsequent chronic complications [6]. Therefore, the diagnosis and treatment of hepatic IRI require the exploration of additional molecular targets.

Cell death is a key pathological mechanism in hepatic IRI. Multiple death modalities interact to determine the injury's severity and progression [8]. During IRI, apoptosis, necroptosis, and pyroptosis exhibit extensive molecular crosstalk and dynamically interact to shape cellular injury and tissue damage [9]. Oxygen deprivation and adenosine triphosphate (ATP) depletion initiate reactive oxygen species (ROS) generation and disrupt calcium homeostasis, triggering initial cell death mechanisms. Paradoxically, reperfusion—the restoration of oxygen supply—exacerbates injury by activating inflammatory cascades and promoting further cell death [10–12]. These interconnected cell death pathways not only directly result in hepatocyte loss but also amplify inflammatory responses through damage-associated molecular patterns (DAMPs), creating a cycle that determines the severity and prognosis of hepatic ischemia-reperfusion injury [13].

PANoptosis is a recently described programmed cell death pathway characterized by the integrated activation of apoptosis, necroptosis, and pyroptosis [14]. This coordinated cell death is executed by a multiprotein complex known as the PANoptosome [15–20]. Assembly of the PANoptosome is triggered by diverse cellular stresses or pathogenic stimuli, with key upstream sensors -including AIM2, RIPK1, ZBP1 and NLRP12 – nucleating distinct complexes: the AIM2-PANoptosome in response to herpes simplex virus 1 and *Francisella novicida* [21]; the RIPK1-PANoptosome during Yersinia infection [22]; theZBP1-PANoptosome in influenza A virus infection infection [23]; and the NLRP12-PANoptosome in hemolytic diseases such as malaria and sickle cell disease [24]. The discovery of these context-specific PANoptosomes underscores the role of PANoptosis in host defense and disease pathogenesis, revealing potential therapeutic targets for a range of conditions.

Growing evidence indicates that PANoptosis plays critical roles beyond infectious diseases, including in tumors and inflammatory conditions [18,25–29]. Karki et al. demonstrated that IRF1-mediated PANoptosis reduces colorectal tumor development in a mouse model of colitis-associated tumorigenesis [30]. Similarly, pyroptosis, apoptosis, and necroptosis mediate myocardial IRI in both preclinical and clinical models [31]. Hepatic IRI, a form of sterile inflammation, shares features with other inflammatory conditions involving PANoptosis. Although direct evidence linking PANoptosis to hepatic IRI is lacking, apoptosis, pyroptosis, and necroptosis have each been implicated as mediators of this injury. The involvement of multiple cell death pathways in IRI-induced liver injury suggests a potential role for PANoptosis. Given that hepatic IRI remains a clinical challenge in liver surgery, identifying PANoptosis-related genes may elucidate underlying mechanisms and reveal novel therapeutic opportunities. Here, we integrated machine learning and bioinformatics to identify PANoptosis-related genes in hepatic IRI and validated these findings experimentally. Our findings advance the understanding of programmed cell death in hepatic IRI and reveal potential therapeutic and diagnostic targets.

## Materials and methods

### Data acquisition and panoptosis gene set establishment

Seven hepatic IRI datasets (GSE151648, GSE12720, GSE15480, GSE23649, GSE87487, GSE112713, and GSE189539) were obtained from the Gene Expression Omnibus (GEO) (S1 Table). Gene symbols were obtained by converting probe names with platform annotation files. Data were $\log_2$-transformed and normalized using the limma package in R. Differentially expressed genes (DEGs) were identified based on adjusted $P$-values and absolute $\log_2$ fold-change (FC) using the criteria $P < 0.05$ and $|FC| > 1$. A PANoptosis pathway gene set was established by integrating genes from cell death-related pathways in the Molecular Signatures Database (MSigDB) (S2 Table).

### Gene Set Enrichment Analysis (GSEA)

GSEA was performed to determine whether the PANoptosis gene set was differentially regulated between hepatic IRI and control samples. Gene expression data were normalized and $\log_2$-transformed. Genes were ranked based on their $\log_2$ fold change between groups, and enrichment scores (ES) were calculated. The ES quantifies the degree to which genes are overrepresented at the top or bottom of the ranked list. Statistical significance was assessed through permutation testing, and $P$-values were calculated by comparing observed ES to the null distribution. False discovery rate (FDR) correction was applied, with gene sets at $FDR < 0.05$ considered significantly enriched.

### Protein-Protein Interaction (PPI)

STRING (version 12.0) was used to assess interactions among PANoptosis-related genes. Cytoscape (version 3.10.2) was employed for PPI network building and visualization. GeneMANIA database was employed to further investigate relationships among important genes.

## Analysis of functional enrichment for PANoptosis-related DEGs

To investigate biological processes (BP), cellular components (CC), molecular functions (MF), and potential pathways, we utilized the "clusterprofiler" R package to GO and KEGG enrichment analysis on DE-NRGs. The $p < 0.05$ was considered significant.

## Hepatic IRI subtype characterization and visualization

To delineate molecular subtypes within hepatic IRI, we employed the Multi-Omics integration and VIsualization in Cancer Subtyping (MOVICS) package, a validated tool for multi-omics data integration in cancer subtyping. Initially, the getClustNum function within MOVICS was leveraged to estimate the optimal number of subtype clusters. This function employs an integrated approach, considering the Clustering Prediction Index (CPI), Gaps-statistics, and Silhouette score to provide a robust estimation of clustering validity across different potential cluster numbers. Based on the synthesis of these metrics calculated by getClustNum, we determined that a three-subtype classification was most appropriate for our dataset.

Subsequently, we inputted a methods list comprising ten distinct and complementary clustering algorithms (COCA, IntNMF, NEMO, moCluster, PINSPlus, iClusterBayes, SNF, ConsensusClustering, CIMLR, and LRA). These algorithms were strategically chosen to represent a diverse range of computational approaches for multi-omics integration, including matrix factorization (e.g., IntNMF, iClusterBayes), similarity network fusion (e.g., SNF, NEMO), consensus strategies (e.g., COCA, ConsensusClustering), and Bayesian frameworks (e.g., iClusterBayes). This consensus approach leverages the collective strength and overcomes the limitations of any single algorithm, thereby defining highly confident and stable hepatic IRI subtypes. [32].

## Machine learning to screen for signature genes

We combined eleven different machine learning techniques and ultimately evaluated 83 technique combinations. The integrated machine learning methods encompassed Partial Least Squares Regression for Generalized Linear Models (plsRglm), Boosted Generalized Linear Models (glmBoost), Ridge Regression (Ridge), the Elastic Net (Enet), the Least Absolute Shrinkage and Selection Operator (Lasso), Gradient Boosting Machine (GBM), eXtreme Gradient Boosting (XGBoost), Support Vector Machines (SVM), Stepwise Generalized Linear Model (Stepglm), Linear Discriminant Analysis (LDA), and the Naive Bayes Classifier (NaiveBayes). This selection provided a diverse methodological foundation, incorporating regularized regressions, boosting ensembles, support vector machines, probabilistic classifiers, and generalized linear model variants. We adopted a sequential approach, in which a diagnostic model was established using the GSE151648 dataset, on five external independent datasets (GSE12720, GSE15480, GSE23649, SE87487, GSE112713). To evaluate model selection, we will calculate the Harrell consistency index (C-index). The model that displays the highest average C-index across all queues will be identified as the optimal model.

## Immune analysis

CIBERSORT [33], Single-sample GSEA (ssGSEA) was implemented to compute scores for infiltrating immune cells and to assess immune functions. To visualize the associations, correlation heatmaps and data plots (created using the "corrplot" package) were generated. These visualizations depicted the connections between infiltrating immune cells and immune function [34].

## Single cell

A total of 75,231 cells from tissue samples of liver were used for this study (GSE189539). The "Seurat" R package was applied for conducting certain key functions such as quality control, statistical calculations, and broad data exploration. Dimension reduction was accomplished using the t-distributed stochastic neighbor embedding (t-SNE) algorithm, along

with the initial principal components (PCs), to facilitate the classification of all cell groups. Utilizing the "limma" R package, we identified significant marker genes within each cluster. The "SingleR" package also provided cell cluster annotation and discrimination.

### Animal and hepatic IRI model

Twelve male C57BL/6 mice, aged 8–10 weeks, were obtained from the Animal Experimentation Center of Xinjiang Medical University (Xinjiang, China; six per group). Housed in a specific-pathogen-free environment, the mice were maintained at a stable temperature of around 21°C. Mice were maintained on a food restriction schedule before treatment, and free access to water was provided for 8 hours. Anesthesia was achieved through intraperitoneal administration of pentobarbital sodium (1%, 50 mg/kg). Segmental (70%) warm ischemic damage was induced by the temporary occlusion of perfusion to the left lateral and median lobes by a vascular clamp. Ischemia was maintained for 60 minutes, and then the vascular clamp was released, and reperfusion was permitted for 6 hours [36]. For the sham group, procedural intervention was merely exposing the hepatic portal vein with no ischemia induced. Following a 6-hour reperfusion period, blood and liver tissue samples were collected. Animals were euthanized under pentobarbital sodium anesthesia by cardiac blood withdrawal followed by cervical dislocation. All procedures were conducted in accordance with ethical approval (IACUC-JT-20240711–19).

### Hepatocellular function detecting and Hematoxylin-Eosin staining

Serum from each group was centrifuged and, following the manufacturer's protocol (Mindray, Shenzhen, China), then analyzed biochemically to evaluate changes in liver function markers (sALT, sAST, and sLDH). Concurrently, the prepared paraffin blocks were sectioned at a thickness of 5 μm, and the paraffin sections were stained with hematoxylin and eosin (H&E) (Biosharp, China).

### Real-time quantitative fluorescence PCR (qRT-PCR)

Employing the TransZol Up Plus Kit (TransGen, Cat: ER501), total RNA was extracted from the samples. The HiScript kit (Vazyme, Cat: R323-01) was used for reverse transcription. Gene expression levels were determined by qPCR using ChamQ (Vazyme, Cat: Q311-02) and normalized to GAPDH expression levels ($2-\Delta\Delta Ct$ method). Gene-specific primer pairs are documented in S3 Table.

### Statistical analysis

Hiplot Pro (https://hiplot.com.cn/), a comprehensive web service for biomedical data analysis and visual interpretation, was used to perform correlation analysis using its integrated Corrplot tool. Statistical analysis was performed by evaluating continuous variables through mean and standard error, utilizing either Student's t-test or Wilcoxon rank sum test for comparative assessments. Statistical analyses were performed using R (version 4.2.1) and GraphPad Prism 9.5. Statistical significance was determined by p values, with *$P<0.05$, **$P<0.01$, ***$P<0.001$, and ****$P<0.0001$, denoting progressively more robust statistical evidence.

## Result

### Identification and enrichment analysis of DE-PANRGs

The overall design of this study is depicted in Fig 1. We assembled pathway gene sets, which included REACTOME_PYROPTOSIS (27 genes), HALLMARK_APOPTOSIS (161 genes), KEGG_APOPTOSIS (87 genes), REACTOME_APOPTOSIS (179 genes), in addition to KEGG_NECROPTOSIS (160 genes), ultimately forming a PANoptosis-related gene (PANRG) set of 485 genes. We performed GSEA on these pathways in the ischemia-reperfusion injury dataset,

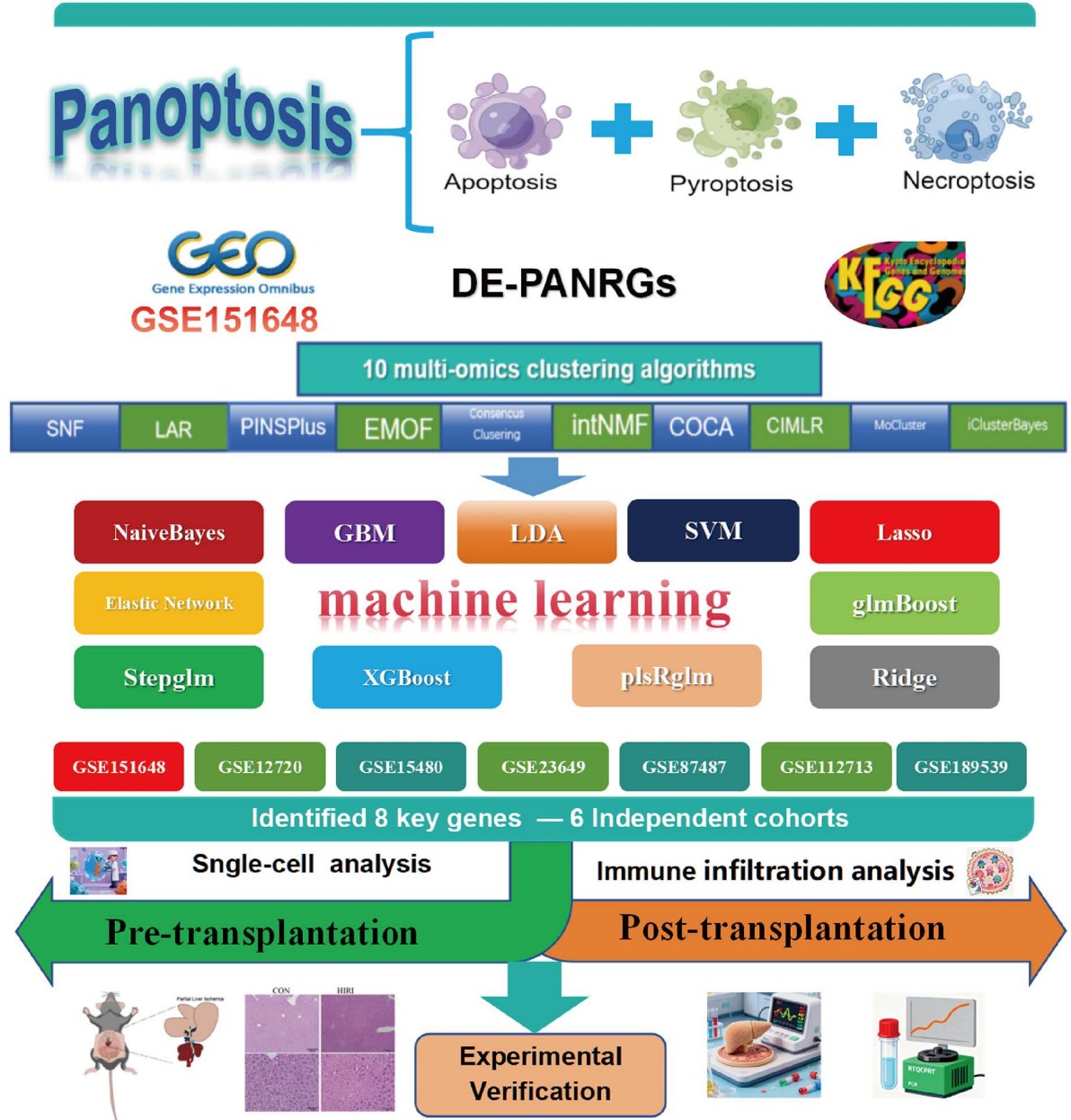

**Fig 1. Overview of study workflow.** The study established a Panoptosis gene set by integrating apoptosis, pyroptosis, and necroptosis-related genes. Using the hepatic IRI-related dataset GSE151648 from GEO, 52 DE-PANRGs were identified, followed by GO and KEGG enrichment analyses. Subsequently, ten multi-omics clustering algorithms were applied to identify molecular subtypes. Based on this, a predictive model was constructed using 11 machine learning algorithms to screen for 8 key genes. These genes were rigorously validated in six independent cohorts, followed by single-cell analysis, immune infiltration analysis, and experimental verification. *GEO, Gene Expression Omnibus; KEGG, Kyoto Encyclopedia of Genes and Genomes; GO, Gene Ontology; CON, Control; IRI, Ischemia-Reperfusion Injury; DE-PANRGs, Differentially Expressed Panoptosis-Related Genes.*

revealing significant enrichment. Notably, the pathways revealed enrichment characterized by these p-values: REAC-TOME_PYROPTOSIS ($p = 0.008646$), PANoptosis ($p = 1e\text{-}10$), HALLMARK_APOPTOSIS ($p = 1e\text{-}10$), REACTOME_APOPTOSIS ($p = 0.0186$), KEGG_NECROPTOSIS ($p = 6.573e\text{-}06$), and KEGG_APOPTOSIS ($p = 0.002508$) (Fig 2A).

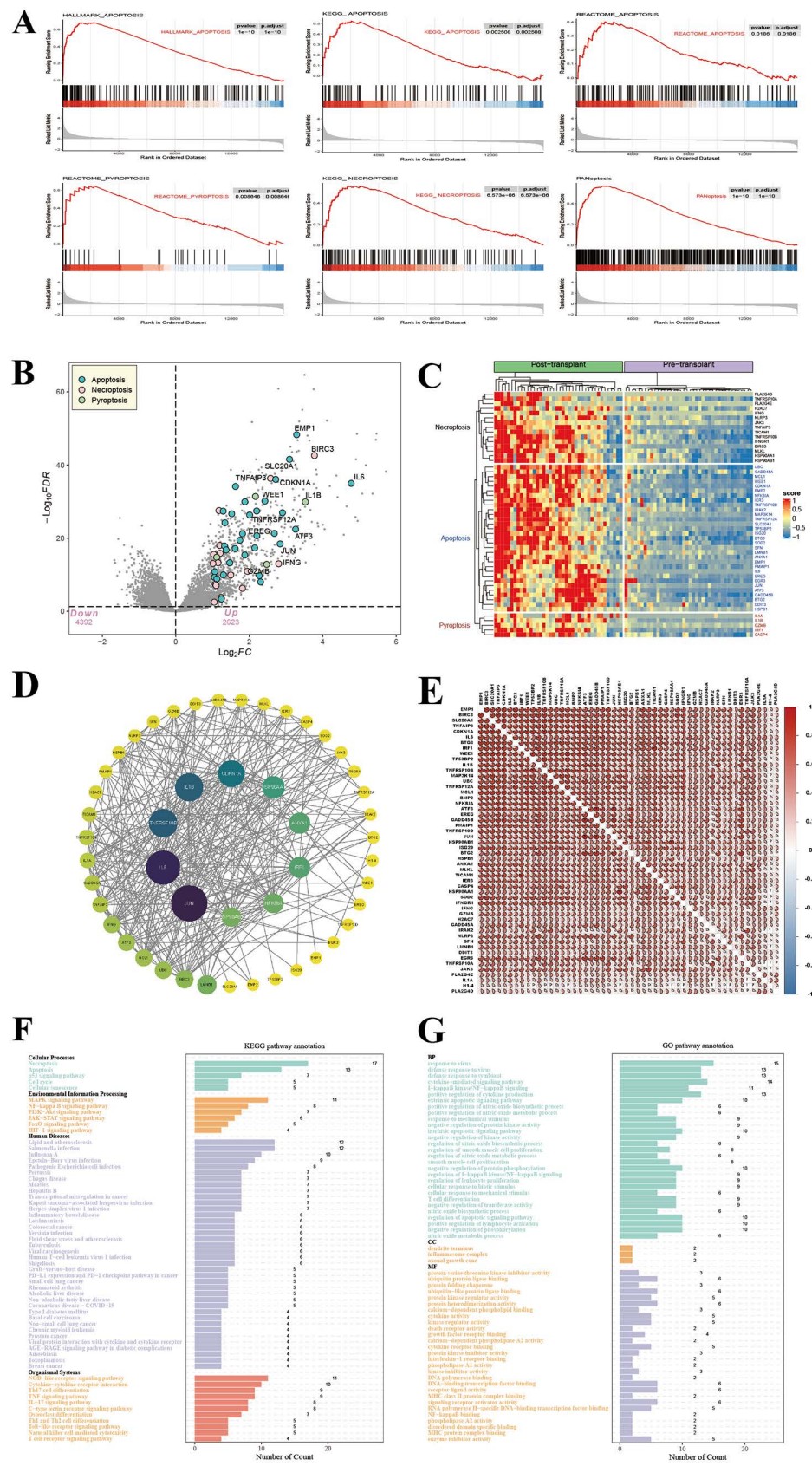

**Fig 2. Identification and enrichment analysis of DE-PANRGs. (A)** GSEA enrichment plot highlights differential enrichment of 'REACTOME_PYRO-PTOSIS', 'PANoptosis', 'HALLMARK_APOPTOSIS', 'REACTOME_APOPTOSIS', 'KEGG_NECROPTOSIS', and 'KEGG_APOPTOSIS' pathways in pre-versus post-transplantation samples. **(B)** The volcano plot shows differentially expressed panoptosis genes. The labeled genes reflect the differences between cell death pathways including apoptosis, necrosis and pyroptosis. **(C)** The heatmap shows 52 DE-PANRs. Red indicates high expression; blue indicates low expression. **(D)** The protein-protein interaction network of 52 DE-PANRs. **(E)** Correlation heatmap of 52 DE-PANRs. **(F)** KEGG and **(G)** GO enrichment analysis of 52 DE-PANRs.

Differential expression analysis of the GSE151648 dataset (adjusted $P < 0.05$ and $|FC| > 1$) identified 946 differentially expressed genes (DEGs). Intersection with the PANRG set yielded 52 differentially expressed PANoptosis-related genes (DE-PANRGs) (Fig 2B). A heatmap displayed the expression patterns of these 52 DE-PANRGs between sham and hepatic IRI samples (Fig 2C). Correlation and protein-protein interaction (PPI) network analyses revealed interactions among the DE-PANRGs (Figs 2D–2E). Functional characterization of the 52 DE-PANRGs was achieved through GO and KEGG enrichment analyses. KEGG pathway analysis, organized into four primary domains encompassing cellular processes and human diseases, identified 60 pathways with adjusted $P < 0.05$ and z-score $> 2$ (Fig 2F). The GO analysis presented the top 60 pathways. The GO analysis primarily focused on BP, including regulation of transcription factor activity, regulation of signaling pathways, immune responses, and inflammatory regulation (Fig 2G). KEGG analysis primarily concentrated on pathways related to cell death, immunity and inflammation, like Necroptosis, Apoptosis, NOD-like receptor signaling pathway, MAPK signaling pathway, TNF signaling pathway, IL-17 signaling pathway, and NF-kappa B signaling pathway.

### Identification of molecular subtypes based on DE-PANRGs

Through application of MOVICS, we had three different clustering subtypes (CS). This determination of number of subtypes was made based on a complete analysis of, silhouette score, gap statistic, and cluster prediction index as well as relevant previous research studies (Figs 3B–3C). In addition to that, we merged the results of clustering with unique molecular expression characteristics of transcriptomics, i.e., Apoptosis, Pyroptosis, and Necroptosis, by means of consensus ensemble methodology (Figs 3A,3D–3E). Notably, in hepatic IRI, CS2 had much higher D E-PANRG expression than any other subtype that was tested.

### Analysis of molecular subtype binding pathway and immune infiltration

Currently, MOVICS molecular characterization methods primarily classify molecular expression levels, potentially associated with specific biological functions. Consequently, our research aimed to investigate the different molecular characteristics underlying hepatic IRI. ssGSEA was employed to evaluate molecular signatures across different groups. Notably, significant enrichment of inflammatory, immune, stress, and metabolism-related pathways emerged in C2, indicating its potential alignment with the prevalent inflammatory phenotype. Different subtypes may be more suitable for specific treatments (Fig 4A). We characterized immune cell infiltration patterns, revealing that immune infiltration was most pronounced in group2, followed by group3, with the lowest in C1. Five types of immune cells (Central memory CD4 T cell, Eosinophil, and Immature dendritic cell, among others) exhibited notable variability across molecular subtype classifications (Fig 4B–4C).

### Screening signature genes by machine learning

Utilizing ten machine learning approaches, we screened key genes associated with hepatic IRI. Leveraging the research cohort GSE151648 and five additional validation datasets from the GEO database (GSE12720, GSE15480, GSE23649, GSE87487, and GSE112713), we systematically validated our findings. We constructed a consensus model by combining ten algorithms. To quantify the model's ability to accurately predict outcomes, we calculated the average C-index for

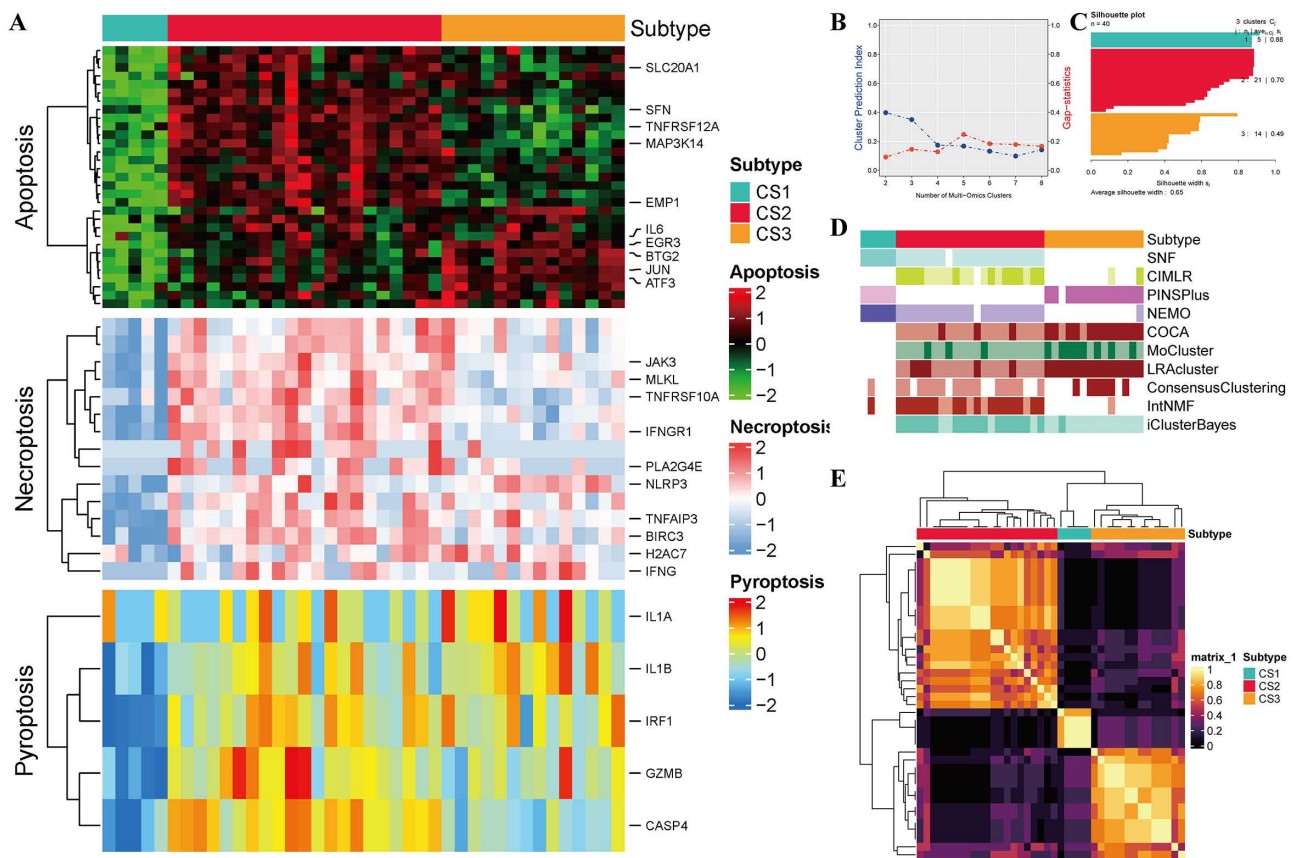

**Fig 3. Identification of molecular subtypes based on DE-PANRGs. (A)** Heatmap displaying gene expression patterns related to apoptotic, necroptotic, and pyroptotic pathways across three distinct molecular subtypes. **(B)** Gap statistical analysis to determine the optimal number of multiomics clusters. **(C)** Sample similarity within each subgroup was evaluated using the Silhouette score. **(D)** Clustering of hepatic IRI patients employing ten advanced multiomics clustering techniques. **(E)** Consensus clustering matrix illustrating three distinct subtypes derived from various clustering algorithms.

each algorithm across all cohorts, a metric indicative of discrimination and predictive accuracy. (Fig 5A–5B). Based on the results, the Lasso+GBM model scored the highest and identified eight key genes: IER3, CDKN1A, EMP1, IL1B, BTG3, JUN, HSPB1, and IL1A. We further evaluated the diagnostic performance of individual genes across these datasets using ROC curves and demonstrated their expression differences through heatmaps and lollipop plots (Fig 5C–5E). GeneMA-NIA analysis of these eight core genes revealed their critical roles in various cellular processes and pathological states (Fig 5F).

## Single-cell transcriptomic profiling of signature genes in hepatic IRI

By leveraging single-cell RNA sequencing dataset GSE189539, we comprehensively explored cellular diversity and gene expression patterns. Using dimensional reduction methods t-SNE and UMAP, we visualized the distribution of different cell types (Fig 6A–6B). Our analysis revealed eight distinct cell populations—endothelial cells, hepatocytes, myeloid cells, NK cells, T cells, tissue stem cells, B cells, and monocytes—exhibiting clear separation in t-SNE and UMAP visualizations. The agreement between these two dimensionality reduction techniques reinforces the confidence in our cell type assignments. We conducted an additional assessment of the distribution and expression profiles of eight identified genes (IER3, CDKN1A, EMP1, IL1B, BTG3, JUN, HSPB1, and IL1A) across different cell types (Fig 6C–6D). Our findings indicated that

**A**

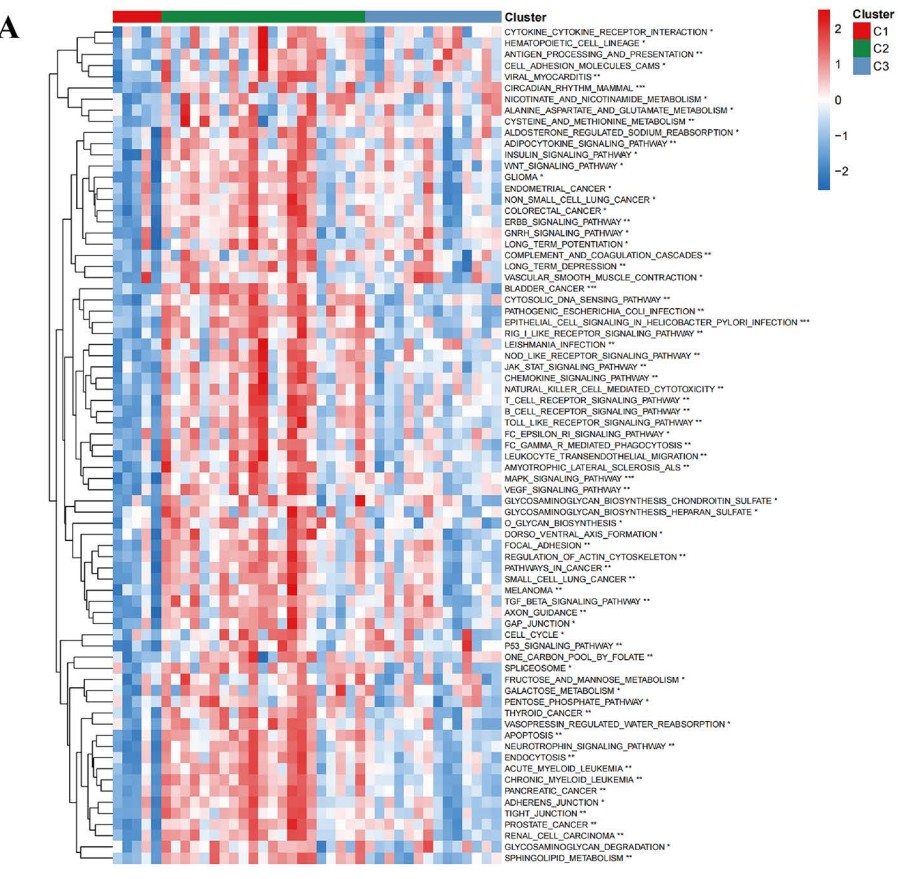

**B**

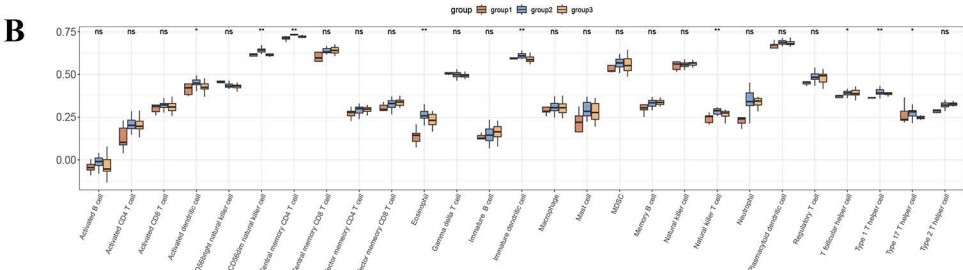

**C**

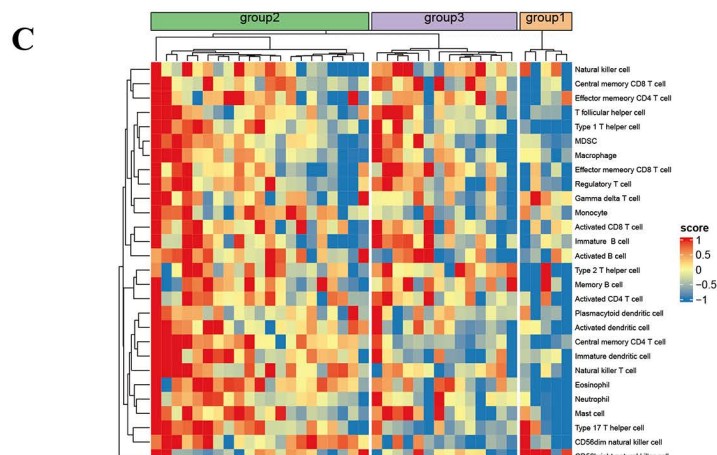

**Fig 4. Analysis of molecular subtype binding pathway and immune infiltration. (A)** ssGSEA highlights pathway activity differences between subgroups. **(B)** and **(C)** Box plots and heatmap analyses reveal differential immune infiltration patterns of 28 immune cell types across three molecular subtypes. *$p < 0.05$; **$p < 0.01$; ***$p < 0.001$; ns, no statistical significance.

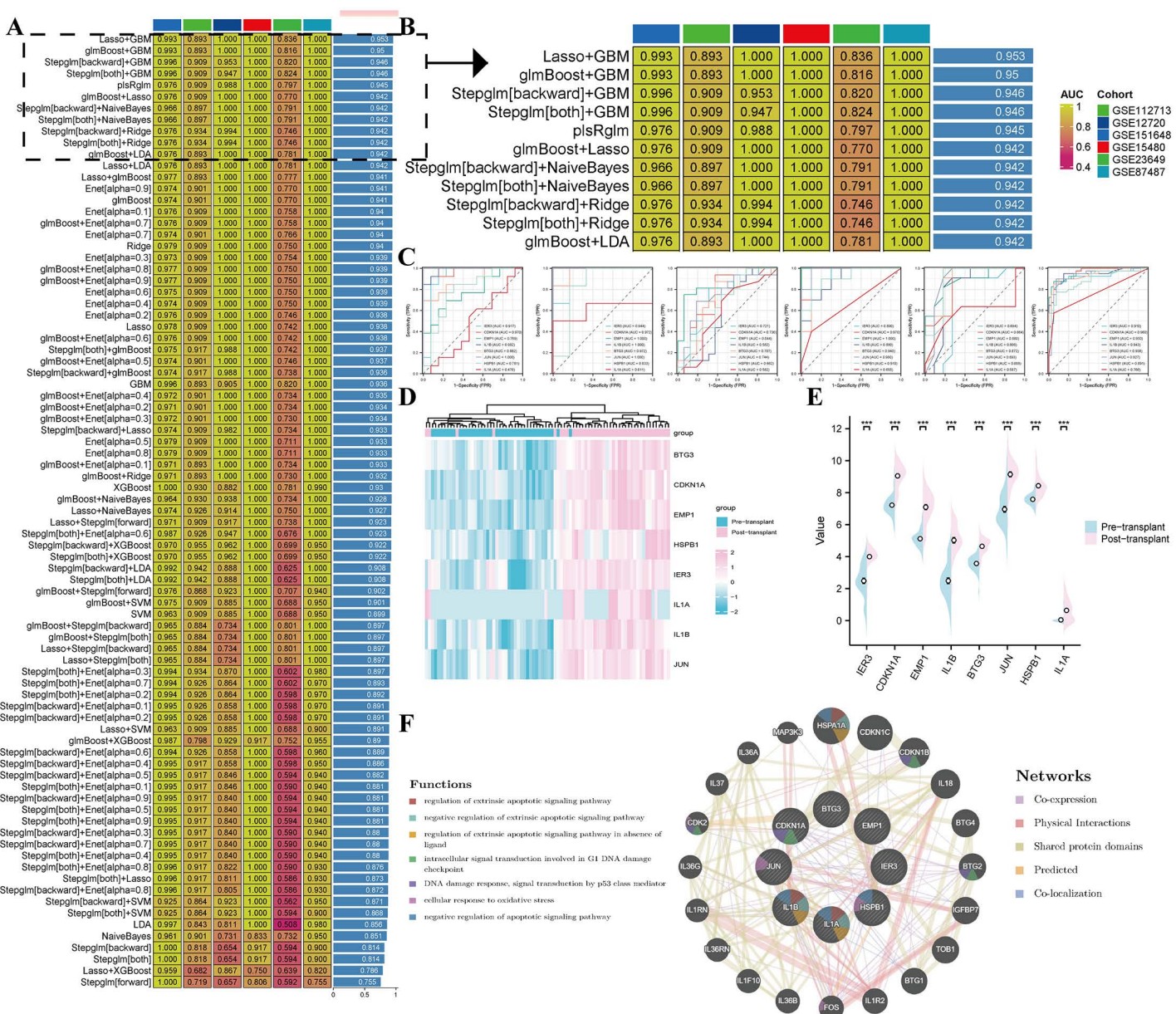

**Fig 5. Screening signature genes by machine learning. (A)** comprehensive computational framework integrating 10 distinct machine learning algorithms was established, with ranked according to the mean C-index. **(B)** Visualization of the performance metrics for the top 11 algorithm combinations across multiple datasets. **(C)** ROC curve analysis of signature genes across 6 independent datasets. Signature genes Pre and Post transplantation gene expression **(D)** Heatmap and **(E)** Violin plots. **(F)** Network analysis of the eight signature genes was performed using the GeneMANIA database.

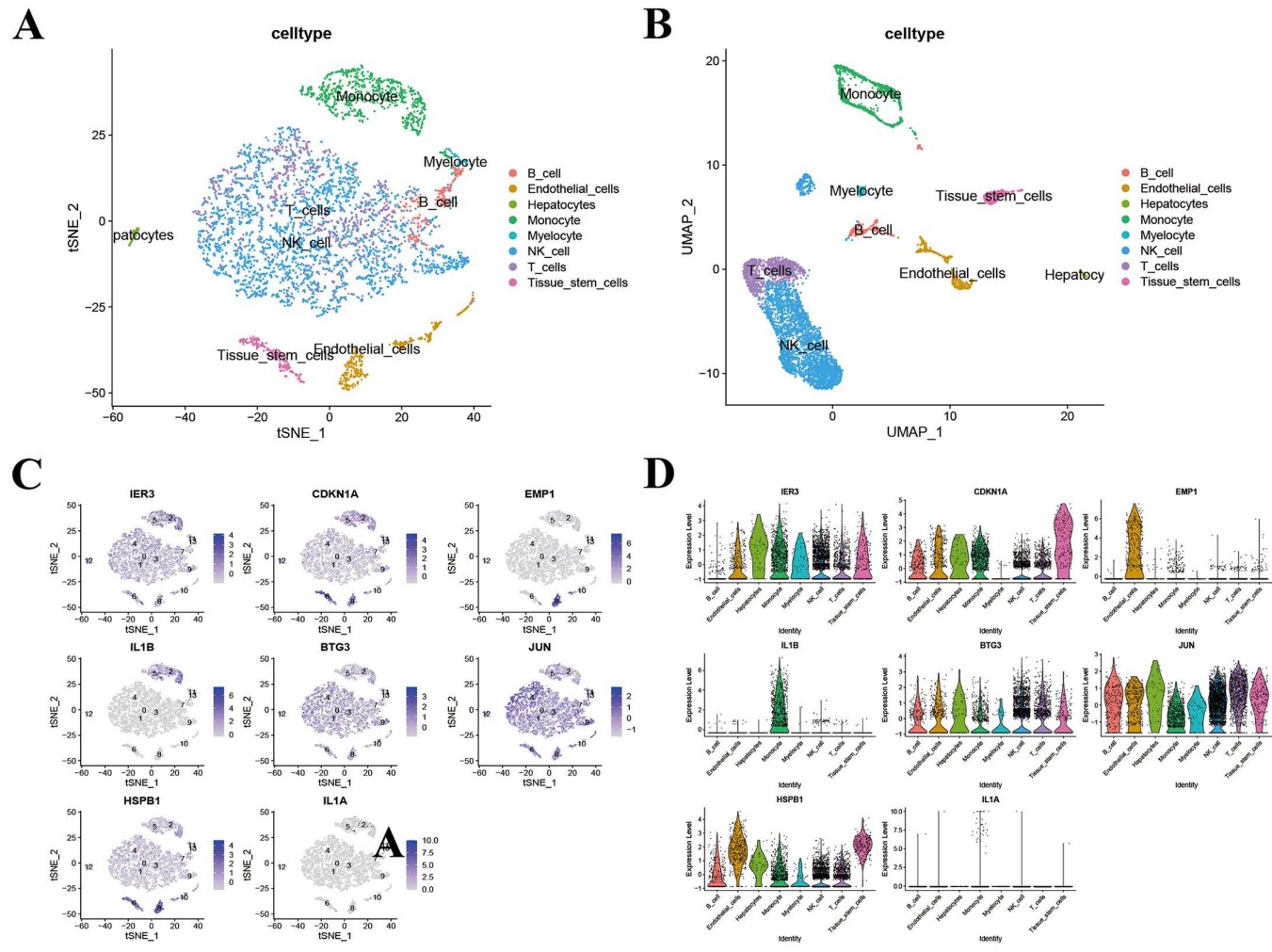

**Fig 6. Single-Cell Transcriptomic Profiling of Signature Genes in Hepatic IRI. (A)** tSNE and **(B)** UMAP analysis of single-cell samples. **(C)** t-SNE plots showing the expression distribution of 8 signature genes at single-cell resolution. **(D)** Feature and violin plots showing the distribution of signature genes in various cell types.

IER3 exhibited ubiquitous expression across all eight cell types, with pronounced levels in monocytes, NK cells, T cells, hepatocytes, and endothelial cells, contrasting with diminished expression in B cells. CDKN1A exhibited minimal expression in myeloid cells, while demonstrating elevated expression in both tissue stem cells and monocytes. EMP1 exhibited a high level of expression in endothelial cells. IL1B showed elevated expression in monocytes. BTG3 was abundantly expressed in NK cells and T cells. JUN's expression was detected in all cell types, with relatively high average levels observed in T cells and hepatocytes. HSPB1's expression was also found in all cell types, displaying particularly high average levels in tissue stem cells, endothelial cells, and hepatocytes. IL1A also showed high expression in monocytes. These results suggest that these genes, with their distinct expression patterns, participate in and play key roles in hepatic IRI.

### Complete assessment of immune cell infiltration signatures in Hepatic IRI

Using CIBERSORT and ssGSEA algorithms, we analyzed immune cell infiltration in liver transplantation samples before and after transplantation, and evaluated the relationship of key genes with immune cell presence. The CIBERSORT tool

was applied to determine the composition of 22 immune cell populations and to assess changes in immune cell infiltration profiles before and after transplantation (Fig 7A–7B). We observed that Macrophages M2 were most abundant in post-transplantation samples, with statistically significant variations (P < 0.05) noted for Macrophages M0, Monocytes, Neutrophils, and resting NK cells. Using the ssGSEA algorithm to analyze 28 immune cell infiltrations (Fig 7C–7D), Monocytes had the maximum infiltration value, with Eosinophils, Activated CD4 T cells, Mast cells, Neutrophils, and Type 17 T helper cells showing the most substantial variations. To determine whether the characteristic genes participate in modulating immune equilibrium, we constructed heatmaps and bubble plots to demonstrate the interplay between the signature genes and immune cells. We observed that the characteristic genes generally showed strong correlations with immune cells (Fig 7E–7F). Eosinophils, Activated CD4 T cells and Mast cells were found to be positively correlated with BTG3, JUN, IL1B and IL1A. IER3 and CDKN1A showed positive correlations with CD56dim natural killer cells, Activated CD4 T cells, and Eosinophils. EMP1 was positively correlated with Eosinophils, CD56dim natural killer cells, and Mast cells. HSPB1 showed positive correlations with Mast cells, Type 17 T helper cells, and CD56dim natural killer cells. These findings indicate that the characteristic genes are significantly involved in modulating immune responses during hepatic IRI.

### Validation of signature gene expression in mouse hepatic IRI

To rigorously test our hypotheses, we induced hepatic IRI in mice by occluding blood flow for 1.5h, followed by 6h of reperfusion (Fig 8A). Through HE staining and detection of sALT, sAST and sLDH, we confirmed the occurrence of hepatic IRI (Fig 8B–8C). qPCR verification revealed a significant increase in mRNA relative expression of IER3, CDKN1A, EMP1, IL1B, BTG3, JUN and IL1A in the hepatic IRI group compared with the control (CON) group ($p < 0.05$). These results were consistent with the trends observed in our previous dataset analysis. However, HSPB1, while showing significant differences, exhibited a trend opposite to our previous human dataset analysis, specifically a notable decrease (Fig 8D).

## Discussion

Hepatic IRI is manifested as a paradoxical exacerbation of cellular dysfunction and death upon the restoration of blood supply to previously ischemic hepatic tissue [1,2]. Prior studies have examined apoptosis, necroptosis, and pyroptosis in the context of IRI [14] the presence of these diverse cell death modes in hepatic IRI, integrating these apoptotic modalities within the context of PANoptosis presents promising prospects. Although direct evidence of PANoptosis in hepatic IRI is currently lacking, the hypothesis of its involvement is reasonable, due to the importance of cell death and inflammation to IRI pathogenesis and its known function in other sterile injuries and infections. We first explored the characteristics of PANoptosis in hepatic IRI, attempting to elucidate its occurrence and progression, with the ultimate goal of providing novel perspectives and strategies for the diagnosis, treatment, and prevention of hepatic IRI.

This research elucidated the potential roles of 485 DE-PANRGs in hepatic IRI, leading to the identification of 52 key genes linked to this recently recognized comprehensive cell death process. By integrating bioinformatics and machine learning approaches, we identified eight PANoptosis-related signature genes (IER3, CDKN1A, EMP1, IL1B, BTG3, JUN, HSPB1, and IL1A), and validated these genes using a mouse IRI model. These genes appear to coordinate complex cell death pathways and immune responses in hepatic IRI. Our findings offer fresh perspectives on the pathophysiological mechanisms underlying this sterile inflammatory process while also highlighting potential therapeutic and diagnostic targets.

Our analysis revealed significant enrichment of PANoptosis-related pathways in hepatic IRI, indicating the concurrent activation of multiple cell death modes. We integrated previously independently studied apoptosis, necroptosis, and pyroptosis within the PANoptosis framework. This integration might amplify inflammatory signal cascades and cellular damage, consistent with our enrichment analysis results. We highlighted several inflammatory and immune-related pathways, including NOD-like receptor signaling pathway, IL-17 signaling pathway, TNF/NF-κB pathway, MAPK signaling pathway, and cytokine-cytokine receptor interaction. This convergence mechanism may drive the sterile inflammation, tissue damage, and repair response that defines hepatic IRI.

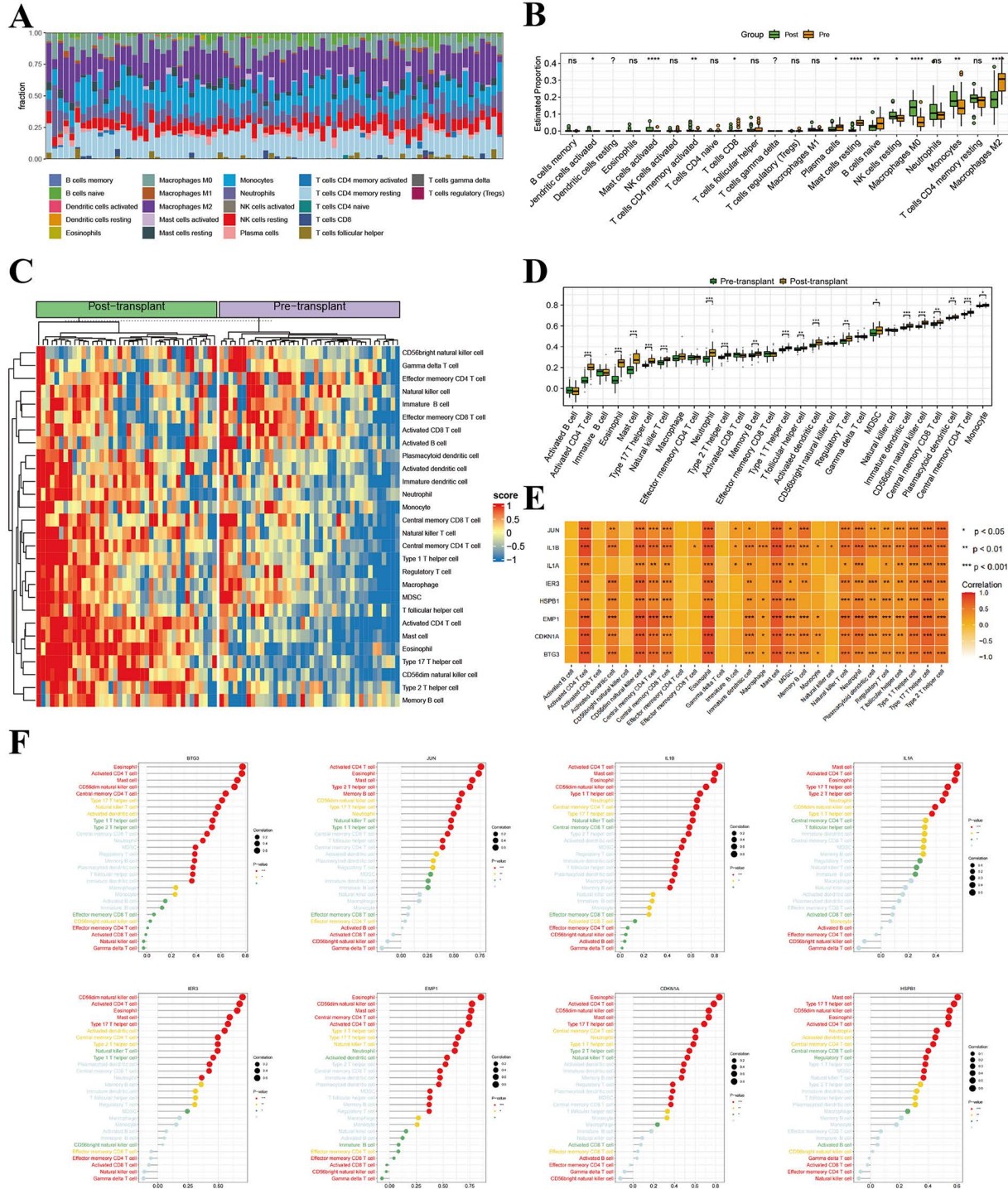

**Fig 7. Complete assessment of immune cell infiltration signatures in Hepatic IRI. (A)** 22 immune cell types in hepatic IRI stacked bar plot showing the relative proportions and **(B)** Box plots comparing the infiltration levels. Abundance profiles of 28 immune cell types **(C)** heat map **(D)** Box plots. Associations between distinct 28 immune cell types (E) matrix heatmap and **(F)** Lollipop plots. ns, no significance; *$p < 0.05$; **$p < 0.01$; ***$p < 0.001$;.

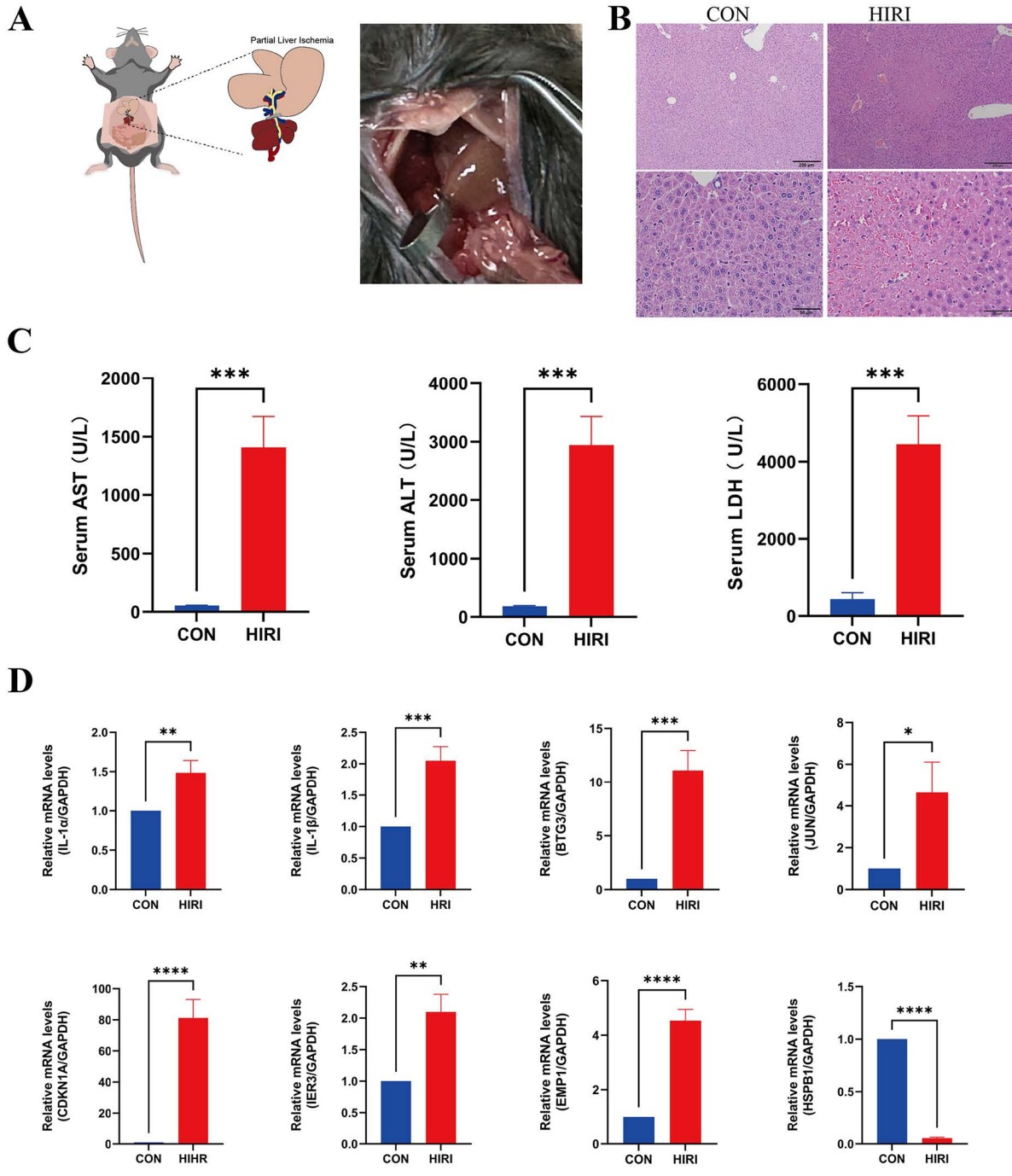

**Fig 8. Validation of Signature Gene Expression in Mouse hepatic IRI. (A)** A Experimental hepatic IRI model. **(B)** Representative images of H&E (200uM, 50uM) of liver-derived from control and hepatic IRI mice 6 h after hepatic IRI. **(C)** sAST, sALT, sLDH in hepatic IRI mice. **(D)** The relative mRNA expression levels of IL-1α, IL-1β, BTG3, JUN, CDKN1A, IER3, EMP1, and HSPB1 in hepatic IRI mice.

Additionally, the algorithm classified hepatic IRI into different molecular subtypes based on DEGs, providing a novel perspective on the heterogeneity of this condition. Our cluster analysis identified three subtypes, with subtype C2 demonstrating high expression of DE-PANRGs. The association of this subtype with inflammatory pathways suggests it may represent a more severe form of hepatic IRI, potentially guiding targeted therapeutic interventions. The correlation between immune cell infiltration and molecular subtypes further emphasizes the importance of the microenvironment in liver injury; significant differences in immune cell populations were observed across different subtypes.

The study indicates that hepatic IRI stimulates activation by recruiting macrophages, dendritic cells, and neutrophils, and activating NK cells, NKT cells, and cytotoxic T cells to exacerbate hepatic IRI [7]. Previous research found that hepatic macrophages (KC) are key cells that aggravate liver injury by producing ROS and secreting pro-inflammatory cytokines; dendritic cells (DC) have a dual role, both promoting liver damage and inhibiting inflammation while facilitating tissue repair. Among lymphocytes, CD4+T cells mediate specific immune responses through inflammatory reactions and differentiation into Th1, Th2, Th17, and regulatory T cell (Treg) subtypes, thereby exacerbating liver damage or regulating inflammation; NK and NKT cells play a bidirectional regulatory role in protecting or damaging the liver through cytokine secretion and interaction with hepatocytes [35,36]. In granulocytes, neutrophils significantly worsen liver injury by releasing ROS, elastase, and forming neutrophil extracellular traps (NETs) [37], while the role of eosinophils remains unclear, though studies suggest they may protect the liver from damage through interactions with macrophages. Through immune infiltration analysis of different molecular subtypes, single-cell analysis, Cibersort, and ssGSEA analysis of liver specimens in two groups, we found significant differences in Dendritic cells, NK cells, T cells, Eosinophils, Macrophages, Monocytes, Neutrophils, and Mast cells.

Based on machine learning analysis, eight differentially expressed PANoptosis-related genes (DE-PANRGs) representing hepatic IRI diagnostic potential were identified. Recognized pro-inflammatory cytokines IL-1α and IL-1β are key mediators of inflammation. IL-1α, released as a bioactive precursor from damaged cells, exerts a significant influence on the course of inflammatory diseases, impacting their onset, progression, and potentially, the subsequent tissue repair processes, in part by modulating the activity of immune cells and resident tissue cells. [38]. IL-1β is associated with multiple cell death modes, with studies reporting that NLRP12 in PANoptosis drives inflammasome formation to promote IL-1β maturation, thereby mediating PANoptosis and inflammatory responses under heme and PAMPs stimulation [24,39]. IER3 is a stress-induced gene regulated by multiple factors, serving as a target gene for NF-κB activation during stress and a prognostic marker [40,41]. CDNK1A (known as p21) is important in protecting tissues against IR injury, upregulated through p53-dependent pathways and activating antioxidant responses by stabilizing Nrf2 protein, thus inhibiting oxidative stress-induced cardiomyocyte death [42]. C-Jun, an essential component of the activator protein-1 (AP-1) complex, is actively involved in diverse cellular functions, including cell growth, programmed cell death, cell survival, and the development of tissues [43,44]. EMP1 (epithelial membrane protein 1) plays a crucial role in various biological processes, such as regulating the cell cycle, aiding in tumorigenesis, and facilitating intercellular communication through signaling pathways [45]. The BTG protein family primarily functions to inhibit cell growth and is involved in advancing the cell cycle and differentiating various cell types [46]. The BTG protein family primarily functions to inhibit cell growth and is involved in advancing the cell cycle and differentiating various cell types [47,48]. We analyzed the expression distribution of these 8 key genes across multiple cell subpopulations at the single-cell level, further supporting their multifunctional characteristics in immune responses, inflammatory signal transduction, and hepatocyte protection/injury processes.

In the mouse IRI model, experimental validation confirmed the upregulation of several key genes (IER3, CDKN1A, EMP1, IL1B, BTG3, JUN, and IL1A) identified through bioinformatics, providing credibility to our findings. Although HSPB1 showed a trend contrary to computational predictions, the differential expression of HSPB1 highlights the complexity of these pathways and emphasizes the importance of further mechanistic research. This discrepancy may be attributed to species-specific regulation, dynamic gene expression time windows, or interactions during the acute injury stage. Studies report that hepatic IRI exhibits significant time-dependent characteristics, and these specific time thresholds provide crucial guidance for constructing clinically relevant experimental model strategies.

Although we employed extensive bioinformatics tools and machine learning algorithms for analysis and conducted preliminary validation in a mouse model, this study has several important limitations. (1) Our analysis based on different datasets from public databases may introduce bias due to differences in sample size, detection platforms, and experimental conditions. (2) Although multiple machine learning algorithms and combinations were evaluated, feature selection prioritized predictive accuracy over mechanistic insight, and the risk of overfitting cannot be completely eliminated. (3) We only conducted transcriptional-level validation in a mouse model without functional assays or protein-level confirmation; species differences may limit human translatability. (4) Clinical data heterogeneity and limited demographic representation restrict subgroup analyses. As a bioinformatics-driven exploratory study, future research will include clinical validation, multi-omics investigations, functional mechanistic studies, and pharmacological testing to bridge the translational gap.

## Conclusion

Ultimately, our investigation illustrates the essential role of PANoptosis in hepatic IRI. Through the integration of bioinformatics, machine learning techniques, and experimental validation, we identified eight signature genes that have the potential to act as biomarkers and therapeutic targets. These findings not only deepen our grasp of the molecular mechanisms involved in hepatic IRI, but also create new prospects for targeted interventions aimed at alleviating liver injury.

## Supporting information

**S1 Table. Information of data applied to this study.** Seven hepatic IRI datasets (GSE151648, GSE12720, GSE15480, GSE23649, GSE87487, GSE112713, and GSE189539) were obtained from the Gene Expression Omnibus (GEO).
(DOCX)

**S2 Table. PANoptosis gene data.** A PANoptosis pathway gene set was established by integrating genes from cell death-related pathways in the Molecular Signatures Database (MSigDB).
(DOCX)

**S3 Table. Gene primer data.** This table lists the forward and reverse primer sequences used for the amplification of specific genes in the study.
(DOCX)

## Acknowledgments

The authors thank the contributors of the public databases used in this study and NCBI for providing the datasets.

## Author contributions

**Data curation:** Pingping Qiao, Talaiti Tuergan, Dalong Zhu, Chang Liu, Rexiati Ruze.

**Funding acquisition:** Tuerganaili Aji, Yingmei Shao.

**Supervision:** Tuerganaili Aji, Yingmei Shao.

**Visualization:** Pingping Qiao, Talaiti Tuergan, Dalong Zhu, Chang Liu, Rexiati Ruze.

**Writing – original draft:** Alimu Tulahong, Xinlu Xu, Aimitaji Abulaiti.

**Writing – review & editing:** Tuerganaili Aji, Yingmei Shao.

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
