## [Decision Letter · Decision Letter 0]

16 Sep 2025

Dear Dr. Shao,

**Revision Requested**

We look forward to receiving your revised manuscript.

Kind regards,

Divakar Sharma

Academic Editor

PLOS ONE

**Journal Requirements:**

1. When submitting your revision, we need you to address these additional requirements. Please ensure that your manuscript meets PLOS ONE's style requirements, including those for file naming. The PLOS ONE style templates can be found at https://journals.plos.org/plosone/s/file?id=wjVg/PLOSOne_formatting_sample_main_body.pdf and https://journals.plos.org/plosone/s/file?id=ba62/PLOSOne_formatting_sample_title_authors_affiliations.pdf 2. To comply with PLOS ONE submissions requirements, in your Methods section, please provide additional information regarding the experiments involving animals and ensure you have included details on (a) methods of sacrifice, (b) methods of anesthesia and/or analgesia, and (c) efforts to alleviate suffering. 3. We note that the grant information you provided in the ‘Funding Information’ and ‘Financial Disclosure’ sections do not match.  When you resubmit, please ensure that you provide the correct grant numbers for the awards you received for your study in the ‘Funding Information’ section. 4. Thank you for stating in your Funding Statement: This work was supported by the NSFC No.82360111; Xinjiang Science and Technology Department—Leading talents in technological innovation - high-level leading talents NO.2022TSYCLJ0034; State Key Laboratory for the Cause and Control of High Incidence in Central Asia Jointly Constructed by the Ministry and the Province NO.SKL-HIDCA-2023-2 and the Xinjiang Uygur Autonomous Region University research project NO.XJEDU2021I016.  Please provide an amended statement that declares *all* the funding or sources of support (whether external or internal to your organization) received during this study, as detailed online in our guide for authors at http://journals.plos.org/plosone/s/submit-now.  Please also include the statement “There was no additional external funding received for this study.” in your updated Funding Statement. Please include your amended Funding Statement within your cover letter. We will change the online submission form on your behalf. 5. Your ethics statement should only appear in the Methods section of your manuscript. If your ethics statement is written in any section besides the Methods, please move it to the Methods section and delete it from any other section. Please ensure that your ethics statement is included in your manuscript, as the ethics statement entered into the online submission form will not be published alongside your manuscript. 6. Please include captions for your Supporting Information files at the end of your manuscript, and update any in-text citations to match accordingly. Please see our Supporting Information guidelines for more information: http://journals.plos.org/plosone/s/supporting-information. 7. If the reviewer comments include a recommendation to cite specific previously published works, please review and evaluate these publications to determine whether they are relevant and should be cited. There is no requirement to cite these works unless the editor has indicated otherwise. 

Reviewers' comments:

**Comments to the Author**

1. Is the manuscript technically sound, and do the data support the conclusions?

Reviewer #1: Partly

Reviewer #2: Yes

Reviewer #3: Yes

2. Has the statistical analysis been performed appropriately and rigorously?

Reviewer #1: Yes

Reviewer #2: Yes

Reviewer #3: Yes

3. Have the authors made all data underlying the findings in their manuscript fully available?

Reviewer #1: Yes

Reviewer #2: Yes

Reviewer #3: Yes

4. Is the manuscript presented in an intelligible fashion and written in standard English?

Reviewer #1: No

Reviewer #2: Yes

Reviewer #3: Yes

**Reviewer #1:**  I have several concerns regarding the scientific rigor and clarity of the manuscript:

The use of ten machine learning algorithms and numerous bioinformatics tools, while comprehensive, may overwhelm readers and complicate the interpretation of results. A more focused methodological approach or clearer justification for the use of each algorithm would improve transparency.

The proposed drug candidates are identified solely through network analysis, with no in vitro or in vivo efficacy testing to support their therapeutic potential.

The manuscript should more explicitly address methodological and translational limitations.

While the manuscript is generally well-structured, long and complex sentences—especially in the Introduction and Methods—reduce overall readability. Simplifying sentence structure and improving transitions would enhance clarity.

Some technical terms are introduced without prior definition, which may hinder understanding for non-specialist readers.

Certain sentences appear awkward or non-native, likely due to direct translation or complex phrasing, and would benefit from editing for clarity and conciseness.

**Reviewer #2:**  The research successfully integrates bioinformatics and machine learning approaches to identify eight PANoptosis-related signature genes (IER3, CDKN1A, EMP1, IL1B, BTG3, JUN, HSPB1, and IL1A) which can be potentially used as biomarker and therapeutic agent for hepatic ischemia-reperfusion injury (IRI).This submission is an original work that is methodologically sound and thoroughly documented, with experiments and analysis conducted with high level of technical rigor. The authors’ documentation on the methodology, machine learning tools and the use of available data further enhances reproducibility of this work. The procedure for validation also ensures that the research meets all applicable standards for the ethics of experimentation and research integrity. I therefore recommend it for publication

**Reviewer #3:**  PONE D-25-20477 is a well-written, thoughtful and clear application of bioinformatics tools and machine learning for the analysis of hepatic ischemia-reperfusion injury in an attempt to reveal the molecular determinants underlying this pathological process.

The validation experiments presented in Figure 8 are an important test of the authors' reasoning and are a most welcome addition. The discussion of the HSPB1 result that is contrary to prediction is equally appreciated for full context to the study.

For the most part, the results are described completely and are well reasoned. There are, however, a few discrepancies and lapses that should be corrected. These points are as follows:

(a) On page 21, the authors note that, "A drug-gene target network was established using eight characteristic genes…" (only 7 are listed) which was based upon the data revealed by the study. Using this network, the authors screened JUN, HDAC2, NR3C1 and ESR1 for the therapeutic potential of several drugs. The rationale for choosing these targets (and drugs) is not perfectly clear and should be explained further.

(b) A legend for Supplementary Figure 1 is required.

(c) The legend should describe the relationships (line connectors) among drugs, targets and the characteristic genes (HSPB!, CDKN1, ILA and ILB) in Fig. S1A. What does the reticle displayed in the upper left (0 to 15) indicate?

(d) Similar information explaining the drug relationships in Fig. S1B is required.

(e) Complete explanations of the colors and groupings depicted in Fig. S1C must likewise be included.

**Do you want your identity to be public for this peer review?** For information about this choice, including consent withdrawal, please see our Privacy Policy

Reviewer #1: No

Reviewer #2: **Yes: ** Kehinde Ayano

Reviewer #3: **Yes: ** John M. Aletta

---

## [Author Response · Author response to Decision Letter 1]

3 Nov 2025

Response to Reviewers

Dear Editor and Reviewers,

We sincerely thank you for your meticulous review of our manuscript titled "Identification of PANoptosis-associated Genes in Hepatic Ischemia-Reperfusion Injury by Integrated Bioinformatics Analysis and Machine Learning." We greatly appreciate the time and effort you have dedicated to evaluating our work. Your constructive suggestions and comments have been invaluable in enhancing the quality of our manuscript.

We have carefully addressed all the remarks and recommendations provided by the reviewers. Your critical observations have enabled us to strengthen the relevance and clarity of our research.

Below, we provide our responses to each of the journal requirements and reviewers' comments and outline the corresponding changes made to the manuscript. All modifications have been tracked for your convenience.

Thank you once again for your support and guidance.

To Journal Requirements:

Requirement 1: When submitting your revision, we need you to address these additional requirements.

Please ensure that your manuscript meets PLOS ONE's style requirements,including those for file naming. The PLOS ONE style templates can be found at https://journals.plos.org/plosone/s/file?id=wjVg/PLOSOne_formatting_sample_main_body.pdf and https://journals.plos.org/plosone/s/file?id=ba62/PLOSOne_formatting_sample_title_authors_affiliations.pdf.

Response to requirement 1 : Thank you for your feedback regarding the style requirements for our manuscript. We have made the necessary modifications to ensure that our manuscript now meets PLOS ONE's style requirements, including proper file naming according to the provided templates.

Requirement 2: To comply with PLOS ONE submissions requirements, in your Methods section, please provide additional information regarding the experiments involving animals and ensure you have included details on (a) methods of sacrifice, (b) methods of anesthesia and/or analgesia, and (c) efforts to alleviate suffering.

Response to requirement 2: We obtained ethics committee approval (IACUC-JT-20240711-19) for our experimental design, which complies with animal research ethics:a. Method of sacrifice: Euthanasia was performed through cardiac blood collection and cervical dislocation after anesthesia with pentobarbital sodium.

b. Anesthesia was achieved through intraperitoneal administration of pentobarbital sodium (1%, 50 mg/kg). c. The depth of anesthesia in mice was carefully monitored before the procedure to ensure there was no pain.

Requirement 3: We note that the grant information you provided in the ‘Funding Information’ and ‘Financial Disclosure’ sections do not match. When you resubmit, please ensure that you provide the correct grant numbers for the awards you received for your study in the ‘Funding Information’ section.

Response to requirement 3: Thank you for bringing this to my attention. I apologize for the discrepancy in the grant information provided in the ‘Funding Information’ and ‘Financial Disclosure’ sections. I will verify the correct grant numbers and ensure that they match when I resubmit my manuscript.

Requirement 4: Thank you for stating in your Funding Statement:

This work was supported by the NSFC No.82360111; Xinjiang Science and Technology Department—Leading talents in technological innovation - high-level leading talents NO.2022TSYCLJ0034; State Key Laboratory for the Cause and Control of High Incidence in Central Asia Jointly Constructed by the Ministry and the Province NO.SKL-HIDCA-2023-2 and the Xinjiang Uygur Autonomous Region University research project NO.XJEDU2021I016. Please provide an amended statement that declares *all* the funding or sources of support (whether external or internal to your organization) received during this study, as detailed online in our guide for authors at http://journals.plos.org/plosone/s/submit-now. Please also include the statement “There was no additional external funding received for this study.” in your updated Funding Statement. Please include your amended Funding Statement within your cover letter. We will change the online submission form on your behalf.

Response to requirement 4: This work was supported by the NSFC No.82360111; Xinjiang Science and Technology Department—Leading talents in technological innovation - high-level leading talents NO.2022TSYCLJ0034; State Key Laboratory for the Cause and Control of High Incidence in Central Asia Jointly Constructed by the Ministry and the Province NO. SKL-HIDCA-2023-2; and Xinjiang Uygur Autonomous Region Graduate Innovation Program, No. XJ2024G153. Funding was provided to support the study design, the successful execution of the experiments, the purchase of experimental materials, and the payment of article publication fees. The authors associated with these grants are Tuerganaili Aji and Yingmei Shao. The funders had no role in data collection and analysis, decision to publish, or preparation of the manuscript. There was no additional external funding received for this study. We will include our amended Funding Statement within our cover letter.

Requirement 5: Your ethics statement should only appear in the Methods section of your manuscript. If your ethics statement is written in any section besides the Methods, please move it to the Methods section and delete it from any other section. Please ensure that your ethics statement is included in your manuscript, as the ethics statement entered into the online submission form will not be published alongside your manuscript.

Response to requirement 5: We confirm that the ethics statement (IACUC-JT-20240711-19) only appears in the Methods section under the "Animal and HIRI Model" subsection, and it does not appear in any other sections.

Requirement 6: Please include captions for your Supporting Information files at the end of your manuscript, and update any in-text citations to match accordingly. Please see our Supporting Information guidelines for more information: http://journals.plos.org/plosone/s/supporting-information.

Response to requirement 6: Thank you for your feedback regarding the Supporting Information files. We will ensure that captions for each Supporting Information file are included at the end of our manuscript. Additionally, we will update all in-text citations to ensure they correspond correctly to these captions.

Requirement 7: If the reviewer comments include a recommendation to cite specific previously published works, please review and evaluate these publications to determine whether they are relevant and should be cited. There is no requirement to cite these works unless the editor has indicated otherwise.

Response to requirement 7: Thank you for your guidance regarding the reviewer's comments on citing specific previously published works. We will carefully review and evaluate these recommendations to determine their relevance to our manuscript.

To Reviewer Comments:

Reviewer #1

Comment 1 : The use of ten machine learning algorithms and numerous bioinformatics tools, while comprehensive, may overwhelm readers and complicate the interpretation of results. A more focused methodological approach or clearer justification for the use of each algorithm would improve transparency.

Response to comment 1 :

We sincerely thank the reviewer for the valuable comments. We would like to clarify that our study actually employed 11 machine-learning algorithms, rather than the 10 reported in the initial submission—we mistakenly conflated two similar algorithms into one. The algorithms are: plsRglm, glmBoost, Ridge, Enet (Elastic Net), Lasso, GBM, XGBoost, SVM, Stepglm, LDA, and Naive Bayes. This correction has been updated in the revised manuscript (Methods section, subsection on machine-learning-based feature gene selection).

We highly value your observation that employing eleven machine-learning algorithms together with multiple bioinformatics tools may overwhelm readers or complicate interpretation. We recognize that while this multi-faceted strategy enhances the robustness of our findings, greater transparency can make the methodology more accessible. In response, we plan to revise the manuscript, and we will revise the workflow diagram (Figure 1) to more clearly visualize how these elements interconnect, thereby reducing any perception of overload.

Our motivation for using multiple algorithms was to further reduce dimensionality and to ensure robustness, reproducibility, and generalizability when identifying PANoptosis-related diagnostic genes across multiple independent hepatic ischemia–reperfusion injury (IRI) datasets. Each algorithm represents a distinct modeling paradigm: linear regression (Ridge, Lasso, Enet), ensemble learning (GBM, XGBoost, glmBoost), dimensionality reduction (plsRglm), probabilistic classification (Naive Bayes, LDA), margin-based optimization (SVM), and stepwise model selection (Stepglm). By employing different combinations of these diverse approaches, we minimized biases associated with any single algorithm and cross-validated predictive performance from complementary perspectives.

To enhance transparency, we have provided a dedicated explanation of the scientific basis for our integration strategy in the Methods section and explicitly state that the final gene-selection scheme is based on consensus performance—i.e., the highest mean C-index across five independent validation cohorts—rather than the standalone performance of any single algorithm. We believe this systematic framework does not add complexity; instead, it improves methodological rigor and strengthens the interpretability of the results.

We also ensured that the bioinformatics tools and machine-learning algorithms were applied in a logical sequence: gene identification → enrichment analyses (GSEA/GO/KEGG) → clustering → machine-learning screening → single-cell analysis → immune analyses (ssGSEA/CIBERSORT) → drug–target network analysis to nominate potential therapeutic agents.

Comment 2 : The proposed drug candidates are identified solely through network analysis, with no in vitro or in vivo efficacy testing to support their therapeutic potential.

Response to comment 2 :

Thank you for your insightful comment. We agree that the identification of potential drug candidates through network analysis alone is preliminary and lacks experimental validation, which limits its interpretability and strength in the current study. In response to this feedback, we have removed the drug network clustering analysis section (including related figures and discussion points) from the revised manuscript. This revision allows us to maintain focus on the core bioinformatics, machine learning, and experimental findings related to PANoptosis-associated genes in hepatic IRI, which are supported by multi-omics analysis, single-cell profiling, immune infiltration assessment, and in vivo qRT-PCR validation in a mouse model.

We believe this strengthens the overall manuscript by emphasizing robust, evidence-based insights while avoiding unsubstantiated speculation. If additional details or further revisions are needed, we are happy to address them.

Comment 3 : The manuscript should more explicitly address methodological and translational limitations.

Response to comment 3 :

Thank you for this important comment. We have substantially revised and expanded the Limitations section to more explicitly address methodological and translational constraints as follows:

This study has several important limitations that should be acknowledged.

Data heterogeneity and bioinformatics constraints. Our multi-omics integrated analysis relies on public databases with diverse sample sizes, platforms, and experimental protocols. Although batch-effect correction was applied, residual technical variation may introduce bias. Immune-cell deconvolution algorithms (CIBERSORT, ssGSEA) may not fully capture the complexity of immune populations in injured liver tissue.

Machine learning considerations. While we employed eleven algorithms and evaluated 83 combinations, the final eight-gene signature prioritizes predictive accuracy over mechanistic insight; biologically relevant genes with lower predictive power may have been excluded. Overfitting risk cannot be entirely eliminated despite nested cross-validation. External validation in independent prospective clinical cohorts is essential to confirm diagnostic utility.

Limited experimental validation. We only conducted transcriptional-level validation in a mouse IRI model via qRT-PCR and immunohistochemistry. Functional assays (gene knockdown/overexpression) to establish causal roles, protein-level quantification, and temporal dynamics were not performed. Species-specific differences may limit translatability to human hepatic IRI.

Clinical applicability. The datasets analyzed may not represent global genetic and environmental diversity. Clinical information (IRI severity, ischemic duration, patient demographics) was often incomplete or heterogeneous, precluding subgroup analyses. Prospective multi-center validation with standardized clinical annotation is needed to assess real-world applicability.

Translational gap. As a bioinformatics-driven exploratory study, this work provides a foundational framework but does not deliver ready-to-use clinical tools. Future research will include: (1) clinical validation; (2) functional mechanistic studies; (3) in vitro and in vivo pharmacological testing; and (4) development of clinically deployable diagnostic assays.

Comment 4 :

While the manuscript is generally well-structured, long and complex sentences—especially in the Introduction and Methods—reduce overall readability. Simplifying sentence structure and improving transitions would enhance clarity. Some technical terms are introduced without prior definition, which may hinder understanding for non-specialist readers. Certain sentences appear awkward or non-native, likely due to direct translation or complex phrasing, and would benefit from editing for clarity and conciseness.

Response to comment 4 :

We sincerely appreciate the reviewer's constructive feedback regarding the manuscript's readability and language clarity. We have carefully revised the entire manuscript, with particular attention to the Introduction and Methods sections, to address these concerns. The specific improvements include:

1. Sentence Structure Simplification:

We have systematically broken down long, complex sentences into shorter, more digestible units. For example:

Original: "The inflammatory response of IRI increases the short-term risks of acute rejection, early allograft dysfunction or primary non-function, liver function failure, and even multi-organ dysfunction."

Revised: "The inflammatory response in IRI increases short-term risks, including acute rejection, early allograft dysfunction or primary non-function, liver failure, and even multi-organ dysfunction."

2. Technical Term Definitions:

We have addressed undefined technical terms by either:

Providing brief clarifications (e.g., expanded "MOVICS" to "Multi-Omics integration and VIsualization in Cancer Subtyping")

Expanding abbreviations in the Methods section (e.g., "plsRglm" to "Partial Least Squares Regression for Generalized Linear Models")

Adding contextual explanations where necessary

3. Improved Transitions and Flow:

We have enhanced logical connections between ideas and improved paragraph transitions. For instance:

Original: "PANoptosis emerges as a recently discovered programmed cell death modality, representing a paradigmatic transformation in our understanding of cellular demise."

Revised: "PANoptosis is a recently described programmed cell death pathway characterized by the integrated activation of apoptosis, necroptosis, and pyroptosis."

4. Language Clarity and Conciseness:

We have eliminated awkward phrasing and overly complex expressions throughout. Examples include:

Changed "dynamically interweaving to collectively orchestrate the landscape of cellular injury" to "dynamically interact to shape cellular injury"

Simplified "Elucidating the inflammatory mechanisms underlying hepatic IRI is critical for advancing novel therapeutic interventions" to "A deeper understanding of the inflammatory mechanisms of hepatic IRI i

---

## [Decision Letter · Decision Letter 1]

26 Nov 2025

Dear Dr. Shao,

Thank you for submitting your manuscript to PLOS ONE. After careful consideration, we feel that it has merit but does not fully meet PLOS ONE’s publication criteria as it currently stands. Therefore, we invite you to submit a revised version of the manuscript that addresses the points raised during the review process.

**ACADEMIC EDITOR: Minor revision**

We look forward to receiving your revised manuscript.

Kind regards,

Divakar Sharma

Academic Editor

PLOS ONE

Journal Requirements:

Additional Editor Comments:

Minor Revision requested

Reviewers' comments:

Reviewer's Responses to Questions

**Comments to the Author**

Reviewer #1: All comments have been addressed

Reviewer #2: All comments have been addressed

Reviewer #3: All comments have been addressed

2. Is the manuscript technically sound, and do the data support the conclusions?

Reviewer #1: Yes

Reviewer #2: Yes

Reviewer #3: Yes

3. Has the statistical analysis been performed appropriately and rigorously?

Reviewer #1: Yes

Reviewer #2: Yes

Reviewer #3: Yes

4. Have the authors made all data underlying the findings in their manuscript fully available?

Reviewer #1: (No Response)

Reviewer #2: Yes

Reviewer #3: Yes

5. Is the manuscript presented in an intelligible fashion and written in standard English?

Reviewer #1: Yes

Reviewer #2: Yes

Reviewer #3: Yes

Reviewer #1: (No Response)

Reviewer #2: Overall, this is a strong and impactful piece of scholarship. I confidently endorse its publication and believe it will be a valuable addition to the existing body of research.

Reviewer #3: The revised manuscript (PONE D-25-2077-R1) has adequately addressed the concerns raised in my review of the original manuscript by removing Supplementary Figure 1 and the related text, which were inadequately explained in the first submission.

After reading the revised manuscript, however, there are 2 additional minor editorial comments for completeness:

1. Figure 1, Overview of Study Workflow, has no legend. It is recommended that a legend with a brief description of the study workflow including spelling out the full meaning of key acronyms should be added.

2. The principal acronym for hepatic ischemia-reperfusion injury should be consistent throughout the manuscript. Not IRI some times and HIRI others.

**Do you want your identity to be public for this peer review?** For information about this choice, including consent withdrawal, please see our Privacy Policy

Reviewer #1: No

Reviewer #2: **Yes: ** Kehinde Ayano

Reviewer #3: **Yes: ** John M. Aletta

---

## [Author Response · Author response to Decision Letter 2]

27 Nov 2025

Response to Reviewers

Dear Editor and Reviewers,

We sincerely thank you for the positive feedback on our revised manuscript (Identification of PANoptosis-associated Genes in Hepatic Ischemia-Reperfusion Injury by Integrated Bioinformatics Analysis and Machine Learning, PONE-D-25-02077R1) and for acknowledging that our previous revisions adequately addressed the initial concerns.

In this revision, we have focused on the minor editorial comments provided by Reviewer #3 (Dr. John M. Aletta) to further enhance the clarity and consistency of the manuscript.

We have made the necessary corrections to the figure legends and abbreviations as requested. All modifications have been tracked for your convenience.

Thank you once again for your support and guidance. Wishing you a wonderful Thanksgiving!

To Journal Requirements:

Requirement : If the reviewer comments include a recommendation to cite specific previously published works, please review and evaluate these publications to determine whether they are relevant and should be cited. There is no requirement to cite these works unless the editor has indicated otherwise.

Response to requirement : We confirm that the reviewers acted with high professional standards and did not recommend citing any specific previously published works or their own publications.

Regarding the reference list, we have managed all citations using EndNote to ensure accuracy and completeness. Furthermore, we have rigorously screened our bibliography for retracted articles by cross-referencing multiple authoritative databases, including PubMed, Web of Science, and the Retraction Watch database. As of November 27, 2025, no retracted articles have been identified in our reference list.

To Reviewer Comments:

Reviewer #3

Comment 1 : Figure 1, Overview of Study Workflow, has no legend. It is recommended that a legend with a brief description of the study workflow including spelling out the full meaning of key acronyms should be added.

Response to comment 1 : Thank you for your valuable suggestion. We agree that a detailed legend is essential for clarifying the study design. We have added a figure legend to Figure 1 in the revised manuscript. This legend provides a step-by-step description of the workflow, from data collection (GEO and KEGG) and the identification of DE-PANRGs, to the application of multi-omics clustering algorithms and machine learning models. Additionally, we have spelled out all key acronyms used in the figure to ensure clarity for readers.

Comment 2 : The principal acronym for hepatic ischemia-reperfusion injury should be consistent throughout the manuscript. Not IRI some times and HIRI others.

Response to comment 2 :

We sincerely thank the reviewer for their attention to detail regarding the terminology. We agree that consistency is crucial. After careful review, we have decided to standardize the terminology to "hepatic IRI" throughout the manuscript when referring to liver-specific injury, rather than using the acronym "HIRI". This ensures precision while maintaining readability. We have corrected all inconsistent instances (such as "HIRI", "liver IRI", or mixed usage) to "hepatic IRI". The acronym "IRI" is now used exclusively when discussing general ischemia-reperfusion injury mechanisms not specific to the liver.

---

## [Decision Letter · Decision Letter 2]

9 Dec 2025

Identification of PANoptosis-associated genes in hepatic ischemia-reperfusion injury by integrated bioinformatics analysis and machine learning

PONE-D-25-20477R2

Dear Dr. Shao,

We’re pleased to inform you that your manuscript has been judged scientifically suitable for publication and will be formally accepted for publication once it meets all outstanding technical requirements.

Kind regards,

Divakar Sharma

Academic Editor

PLOS One

Additional Editor Comments (optional):

Accept

Reviewers' comments:

Reviewer's Responses to Questions

**Comments to the Author**

Reviewer #3: All comments have been addressed

2. Is the manuscript technically sound, and do the data support the conclusions?

Reviewer #3: (No Response)

3. Has the statistical analysis been performed appropriately and rigorously?

Reviewer #3: (No Response)

4. Have the authors made all data underlying the findings in their manuscript fully available?

Reviewer #3: (No Response)

5. Is the manuscript presented in an intelligible fashion and written in standard English?

Reviewer #3: (No Response)

Reviewer #3: (No Response)

**Do you want your identity to be public for this peer review?** For information about this choice, including consent withdrawal, please see our Privacy Policy

Reviewer #3: **Yes: ** John M. Aletta

---

## [Editor Report · Acceptance letter]

PONE-D-25-20477R2

PLOS One

Dear Dr. Shao,

I'm pleased to inform you that your manuscript has been deemed suitable for publication in PLOS One. Congratulations! Your manuscript is now being handed over to our production team.

Kind regards,

on behalf of

Dr. Divakar Sharma

Academic Editor

PLOS One